# Perceptual bias is reduced with longer reaction times during visual discrimination

Ron Dekel[1] & Dov Sagi [1]*

Fast and slow decisions exhibit distinct behavioral properties, such as the presence of decision bias in faster but not slower responses. This dichotomy is currently explained by assuming that distinct cognitive processes map to separate brain mechanisms. Here, we suggest an alternative single-process account based on the stochastic properties of decision processes. Our experimental results show perceptual biases in a variety of tasks (specifically: learned priors, tilt aftereffect, and tilt illusion) that are much reduced with increasing reaction time. To account for this, we consider a simple yet general explanation: prior and noisy decision-related evidence are integrated serially, with evidence and noise accumulating over time (as in the standard drift diffusion model). With time, owing to noise accumulation, the prior effect is predicted to diminish. This illustrates that a clear behavioral separation—presence vs. absence of bias—may reflect a simple stochastic mechanism.

[1] Department of Neurobiology, The Weizmann Institute of Science, Rehovot 7610001, Israel. *email: Dov.Sagi@Weizmann.ac.il

Human decisions are often biased. In visual perception, contextual effects, and prior experience lead to systematic biases in the judgment of objects' properties such as orientation, size, and color[1–4]. Experiments show these biases to be reduced with longer exposure duration[5,6], intuitively explained by a hierarchy of processing levels, with the slower processes, computationally more powerful, providing a more veridical perception[5–7]. Here, we consider an alternative, single process, account.

A powerful idea in the neurosciences is that decision makers, brains included, integrate noisy evidence over time to improve performance[8,9]. Theories adhering to this principle, such as drift diffusion models (DDM[10]), offer remarkable explanatory power, notably predicting human reaction-time in decision tasks[11], and accounting for neuronal activity in brain regions correlated with decision-making[8]. In such integrators, the initial state of accumulation is set by prior evidence favoring (biasing) one decision outcome over others[12], implementing an approximation of Bayes' rule[13]. The resulting response bias is expected to decrease with decision time, due to accumulation of evidence and noise.

The prediction observed and investigated here is that perceptual biases are strong with fast decisions and are much reduced, possibly eliminated, with slow decisions, regardless of stimulus duration. Our experimental results confirm this prediction, and suggest that time-dependent bias is due to temporal accumulation of sensory evidence and noise.

## Results

**Prior dependent bias.** When faced with a difficult visual discrimination task in which one of the objects is more probable, observers tend to choose the more frequent alternative when uncertain. For example, consider a task involving a fine discrimination between two oriented objects (Gabor patches, $\sigma = 0.42°$, $\lambda = 0.3°$, see Methods), slightly tilted from vertical, clockwise (CW), or counter-clockwise (CCW) ($+0.5°$ or $-0.5°$), briefly presented (50 ms). This is a challenging task, in the sense that our observers provided the correct answer on only ~70% of the trials. For example, the "+" stimulus was sometimes reported as having a "−" orientation, showing P(answer + |+) = 0.73, and the "+" response was sometimes provided when the "−" stimulus was presented, showing P(answer + |−) = 0.36 (mean, SEM ≤ 0.04; N = 7 observers). The bias in the task, which is the preference for

responding "+" over "−", independent of stimulus orientation, can be quantified by comparing the sum of these two conditional probabilities to 1 (see Fig. 1). Here, the sum shows 1.09, slightly above 1, indicating a small, statistically non-significant, bias in favor of the "+" response ($t_{(6)} = 1.59$, $P = 0.16$, two-tailed $t$ test). Importantly, increasing the occurrence frequency of the $+0.5°$ stimulus to 75% of the trials during a testing block (80 trials) leads observers to being more likely to provide the "+" response independent of the stimulus presented, leading to P(+|+) = 0.81 and P(+|−) = 0.44. Similarly, reducing the occurrence frequency of $+0.5°$ to 25% of the trials leads to a reduced frequency of "+" responses, showing P(+|+) = 0.66 and P(+|−) = 0.30 (mean, SEM ≤ 0.04; an average change in the sum of the conditional probabilites from the 25 to the 75% priors of M = 0.29 with $t_{(6)} = 5.10$, $P = 0.002$, two-tailed paired $t$ test). Overall, these results illustrate the classic finding that decision bias is modified in accordance with learned task priors[1,2].

Next, we split the decision data into two equal-quantity bins, around the median reaction time (RT, reflecting the time it took the observer to provide a response; this was done separately for each experimental block and for each observer, see the Methods). The results showed that the overall bias measured across all trials reflects a strong bias in the faster trials (M = 0.63, $t_{(6)} = 7.91$, $P = 0.0002$, Fig. 1a) and remarkably no bias in the slower trials (M = −0.03, $t_{(6)} = −0.63$, $P = 0.55$, Fig. 1b).

To better quantify the observation of time-dependent bias, we used a probit scale for the probability measurements, i.e., probabilities were transformed using $z(\cdot)$, the inverse cumulative distribution function of the standard normal distribution (using four equal-quantity bins, see measurements at Fig. 2). This description of behavioral data permits convenient visualization in terms of measures motivated by signal detection theory (SDT) for bias and sensitivity[1]

$$\text{Bias} = z(P(+|s; \text{prior}_1)) - z(P(+|s; \text{prior}_2)). \quad (1)$$

$$c = -0.5 \cdot [z(P(+|+)) + z(P(+|-))]. \quad (2)$$

$$d' = z(P(+|+)) - z(P(+|-)). \quad (3)$$

Specifically, bias (Eq. (1)) is the change in response probability to a fixed stimulus $s$ ("+" or "−") under different priors (as illustrated in Fig. 2). This measure is formally equivalent to the average change in the internal decision criterion, $c$ (Eq. (2), Fig. 3b, individual data at Supplementary Fig. 1, see Methods). Using this measure to quantify the bias between the 25 and the 75% priors, we found a robust prior-dependent bias in fast trials, and almost no bias in slow trials (Fig. 3a). To analyze this effect, we used a linear mixed-effects regression of the average bias (across the two stimuli options) as a function of time bin index (bin 1 to bin 4, see the Methods). This analysis revealed a significant overall bias (intercept term, $t_{(26)} = 10.28$, $P = 2 \times 10^{-10}$ Bonferroni corrected for two multiple comparisons), and importantly a significant reduction in bias with time (slope term, $t_{(26)} = −8.68$, $P = 7 \times 10^{-9}$ corrected, slope estimate ± SE of −0.54 ± 0.06). The reduction in bias was robust, observed on an individual basis (N = 7, Supplementary Fig. 2), with a large effect size (Cohen's d = 3.53, Hedges' g = 3.14, for the pairwise differences), and replicated with a larger sample through the Amazon Mechanical Turk (Fig. 3d, e) (N = 50; statistics at Supplementary Table 1).

The corresponding measure for discrimination sensitivity, $d'$ (Eq. (3)), quantifies the observers' ability to discriminate between the two stimuli under a fixed prior, as illustrated in Fig. 2. This measure showed that changes in sensitivity were small and inconsistent ($d' \approx 1$ across decision times, Figs. 1 and

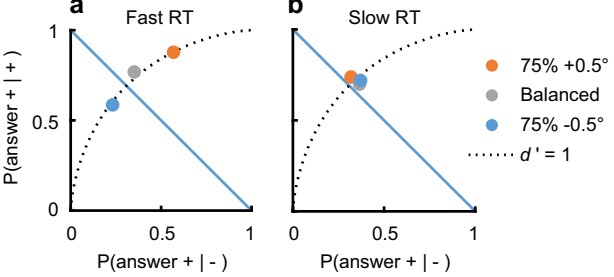

**Fig. 1 Prior-dependent bias disappears with decision time.** Observers (N = 7) discriminated between two oriented visual stimuli with their relative frequency of occurrence varied ("+" stimulus presented in 25, 50, or 75% of trials, with P(−) = 1 − P(+)). Shown is a receiver operating characteristic (ROC) plot depicting P(+|+) as a function of P(+|−), averaged across observers, in reaction times (RTs) that are **a** faster or **b** slower than the median. Data points lying on the dotted curves indicate a discrimination sensitivity of $d' = 1$, with their position on the curve corresponding to decision bias. The negative diagonal, P(+|+) + P(+|−) = 1, represents unbiased performance (see the Methods).

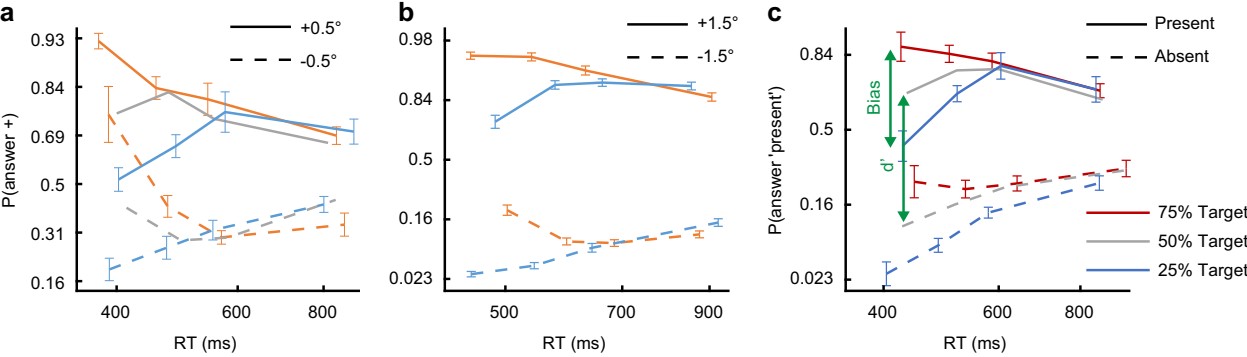

**Fig. 2 Interaction of bias and time for prior-dependent experiments.** Shown is the response probability as a function of RT, in the **a** "Discrimination" ($N = 7$), **b** "Discrimination MTurk" ($N = 50$), and **c** "Detection" ($N = 9$) experiments, for different priors (marked by colors; panels **a**, **b** as in Fig. 1), and different stimuli (line styles), averaged across observers in four equal-quantity bins. Bias (Eq. (1)) is the change in response probability to a fixed stimulus under different priors (vertical distance between data points of different colors). Similarly, $d'$ (Eq. (3)) is the change in response probability under different stimuli for a fixed prior (vertical distance between data points of the same color with different line styles). The results showed a significant prior-dependent bias only in fast responses. Error bars are ±1SEM.

**Fig. 3 Behavioral measures for bias and sensitivity in the prior-dependent experiments.** Shown for the discrimination experiment is the average across observers ($N = 7$) of **a** bias (Eq. (1)) for the different stimuli (line styles), **b** internal criterion ($c$, Eq. (2)) for the different priors (colors), and **c** sensitivity ($d'$, Eq. (3)) also for the different priors, as a function of the average RT, in four equal-quantity bins. **d**–**f** Same for an Amazon Mechanical Turk replication ($N = 50$). **g**–**i** Same for the detection experiment (conducted in the laboratory, $N = 9$). Results showed rapid reduction in bias with time. For individual data see Supplementary Fig. 1. Error bars are ±1SEM.

3c). Specifically, using a linear mixed-effects regression analysis for $d'$ as a function of time bin index (averaged across the two priors, 25 and 75%), we found no significant modulation by time (slope term, $t_{(26)} = 0.26$, $P = 0.8$). The MTurk replication

showed the same, with a statistically significant yet negligible reduction in $d'$ with time ($t_{(198)} = -3.15$, $P = 0.005$ corrected; linear reduction of 14% from the fastest to the slowest bin, Fig. 3f).

Next, we consider a classic detection task, whereby observers report whether a visual target is present or absent from the display (using low-contrast Gabor patches for the target, $\sigma = 0.42°$, $\lambda = 0.3°$). To experimentally manipulate priors, the proportion of target-present relative to target-absent trials was changed (from 25 up to 75%, in different 80-trials blocks). Results showed a significant prior-dependent bias (intercept term, $t_{(34)} = 10.86$, $P = 3 \times 10^{-12}$ corrected, $N = 9$ observers, Fig. 3g, h). Most importantly, the bias was strong in fast decisions, and almost entirely absent in slow decisions (Fig. 3g), showing a significant reduction with time (slope term, $t_{(34)} = -8.89$, $P = 4 \times 10^{-10}$ corrected, slope estimate ±SE of $-0.40 \pm 0.04$). As before, the reduction in bias was clearly observed on an individual basis (Supplementary Fig. 2c), with a large effect size (Cohen's $d = 2.27$, Hedges' $g = 2.07$).

The sensitivity in the task showed an inconsistent but overall significant reduction with time (slope term, $t_{(34)} = -3.15$, $P = 0.003$, Fig. 3i), though not by much (~36% linear reduction from the fastest to the slowest bin). It is interesting to note that the time dependence of the criterion data in the detection task (Eq. (2), Fig. 3h) reflects a tendency to provide an "absent" response in fast responses. In both Detection and Discrimination experiments, RT was mostly stable within block, possibly even slightly increasing, controlling for the potential confound of gradual learning of the prior (Supplementary Fig. 3).

Overall, in both discrimination and detection tasks, a change in behavior due to the learned prior was mostly restricted to the faster half of the responses.

**Context dependent bias**. Motivated by the interaction of decision time and bias when manipulating the prior of the decision alternatives, we were interested in context-dependent biases. By this, we refer to the ubiquitous biases observed for perhaps any visual property (orientation, luminance, color, motion, size, or facial expression[3,4]) due to the contextual value of the same property. For example, a physical vertical target (0°) may be reported as if it is tilted −2° due to a surrounding context of +20° (spatial context, Fig. 4a). The same bias can also be caused by a preceding exposure to a +20° orientation (temporal context, Fig. 4b; similarities of spatial and temporal contexts are discussed elsewhere[14]). Here, we consider how the context-dependent biases interact with RT. Note that the type of context-dependent biases measured here are distinct from the prior-dependent biases analyzed above in the sense that the direction of the bias is opposite to the orientation of the context (hence denoted "-bias"). Moreover, context-dependent biases are clearly at least partially perceptual (e.g., actually seen as reported, thus are referred to as visual illusions), while the question of whether prior-dependent biases are perceptual or decisional is currently under debate (e.g., recently[15]).

First, we measured the influence of oriented context on subsequently perceived orientation (tilt aftereffect, TAE; Fig. 4b, results in Fig. 4d). To verify that the decision time is not confounded with the presentation duration of the target[5,6], we used briefly presented targets (50 ms). Thus, a stimulus sequence consisted of an oriented adapter (+20° or −20° context, 50 ms), followed by a no stimulus interval (600 ms), and a close to vertical target (50 ms). Observers reported the target orientation as being CW or CCW, as in the discrimination experiment reported above. The target orientation that received an equal number of CW and CCW reports was assumed to correspond to the perceived vertical (PV) orientation. The results showed standard TAE magnitudes, with PV biased in a direction opposite to that of the context (raw measurements at Supplementary Fig. 4).

Most importantly, we found that the magnitude of contextual influence (bias as defined in Eq. (1)) was reduced in slower decisions, showing about 50% reduction from the fastest to the slowest bin, as seen in Fig. 4d. To analyze this effect, we applied a linear mixed-effects regression analysis of bias as a function of the time bin index. Results showed a significant overall bias (intercept term, all experiments showing $P < 2 \times 10^{-7}$), and importantly, a significant reduction with time (Fig. 4d) (slope term, fixation: $N = 12$, $t_{(46)} = 6.54$, $P = 9 \times 10^{-8}$ corrected, periphery: $N = 14$, $t_{(54)} = 7.35$, $P = 2 \times 10^{-9}$ corrected, periphery non-retinotopic: $N = 14$, $t_{(54)} = 4.35$, $P = 1 \times 10^{-4}$ corrected; Bonferroni correction for two multiple comparisons; experimental names refer to the retinal positions of the stimuli, with "non-retinotopic" meaning that the adapting context and the target were presented at different retinal positions, and periphery referring to the near-periphery at eccentricity of ±1.8°, see the Methods). The effect was extremely robust, evident across target orientations (Supplementary Fig. 5), on an individual basis (Supplementary Fig. 6), with a large effect size (fixation: Cohen's $d = 2.03$, Hedges' $g = 1.90$, periphery: $d = 1.85$, $g = 1.75$, periphery non-retinotopic: $d = 0.90$, $g = 0.85$). The TAE fixation data from Pinchuk et al.[16] replicated the effect (Fig. 4d, statistics at Supplementary Table 1). Note that unlike the known reduction in the aftereffect magnitude with increased time difference between the adapting context and the target[17], here the involvement of decision mechanisms was measured by analyzing the TAE at different RTs and using a fixed target-to-adapter time, implying a fixed adaptation level.

Next, we measured the influence of surrounding oriented context on the perceived orientation of a central target (tilt illusion, TI, Fig. 4a). The results are shown in Fig. 4c. When the presentation of the stimulus (target + surround) persisted until the observers' response, the results showed a clear reduction in bias for increased RTs ($t_{(38)} = 7.65$, $P = 7 \times 10^{-9}$ corrected, $N = 10$, Fig. 4c), measuring bias of ~3 at ~550 ms which decreased to a bias of ~1.5 at ~2000 ms (effect size: Cohen's $d = 2.78$, Hedges' $g = 2.57$, Supplementary Figs. 4–6). Most importantly, when using a fixed presentation duration (200 ms), the results again showed a reduction in bias with time ($t_{(38)} = 4.23$, $P = 0.0003$ corrected, Fig. 4c), with bias of ~3 at ~500 ms, decreasing to bias of ~2 at ~1000 ms (Cohen's $d = 1.24$, Hedges' $g = 1.14$, Supplementary Figs. 4–6). Therefore, the known reduction in bias for longer presentation durations[5,6,18] can be attributed to changes in decision-related mechanisms that are measured here using RT-based analysis. Findings were replicated using a larger sample size through the Amazon Mechanical Turk ($N > 50$ per experiment, see Fig. 4c; including a "mix" condition pooling data from a number of additional experiments, showing the effect is robust; statistics at Supplementary Table 1).

Unlike bias, the task sensitivity, $d'$, did not show a consistent dependency on RT (as seen in Fig. 5, individual data at Supplementary Fig. 7), suggesting a dissociation between change in bias and change in sensitivity at different RTs (as found for the prior-dependent bias, Figs. 1 and 3). An alternative measure for bias, though less accurate (see Methods) is the shift in the PV orientation measured in degrees. Results using this measure showed similar results as reported above (Supplementary Fig. 8). Overall, the magnitude of both TAE and TI was reduced in slower decisions, revealing an interaction between contextual influence and decision-making processes.

**Theory**. Next, we aimed to account for the observed reduction in bias with decision time by applying general principles. Generally, a system that accumulates noisy evidence when making a decision can be interpreted as a stochastic decision process (e.g.,[19]).

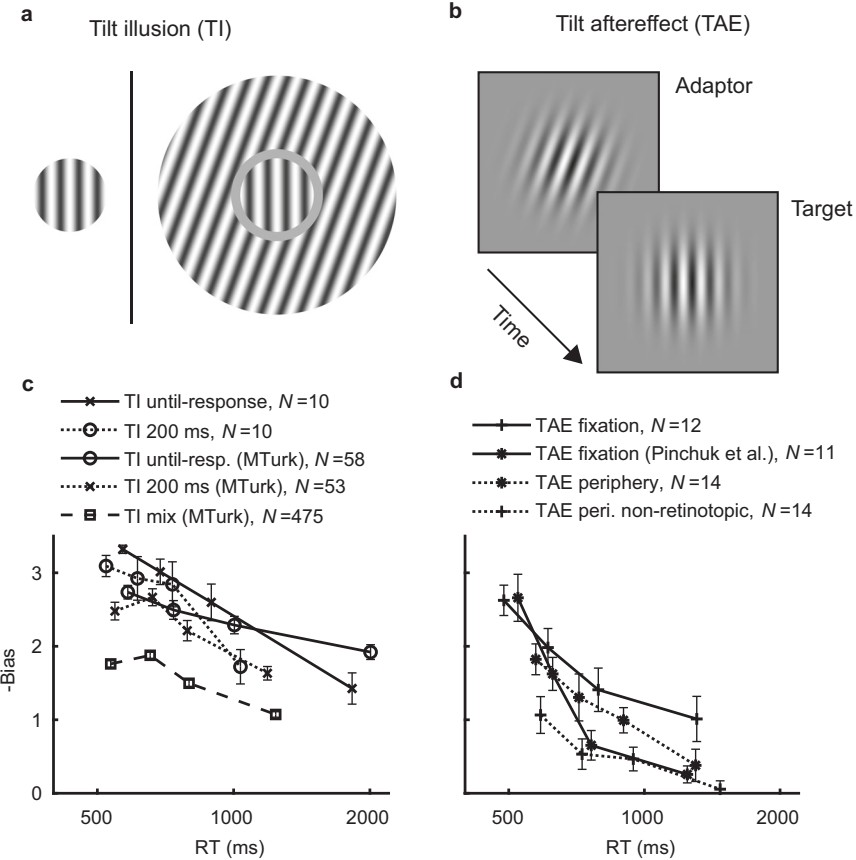

**Fig. 4 Context-dependent bias disappears with decision time. a** Tilt illusion (TI, spatial context). An oriented surrounding context leads to a change in the perceived orientation of a center (left: no surround; right: with surround, like the experimentally used stimuli). **b** Tilt aftereffect (TAE, temporal context). Exposure to an oriented adapter leads to a change in the perceived orientation of a subsequently viewed target. **c** Bias (Eq. (1)) for the TI due to context orientation as a function of reaction time (four bins), for near-vertical targets, averaged across observers. **d** Same, for the TAE. For individual data see Supplementary Fig. 6. Error bars are ± 1SEM.

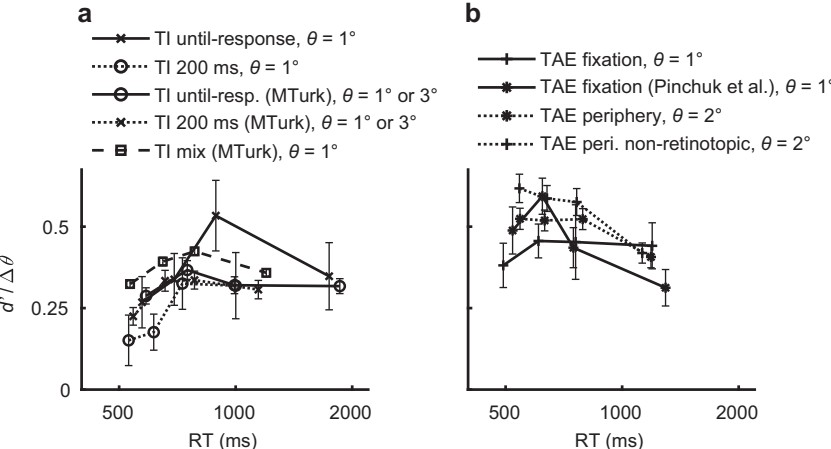

**Fig. 5 Task sensitivity in the context-dependent experiments.** Shown for the **a** TI and **b** TAE experiments is the average across observers of the task sensitivity ($d'$, Eq. (3)) divided by the orientation difference ($\Delta\theta = 2\theta$) as a function of RT (four bins). The $d'$ was measured between the $-\theta°$ and $+\theta°$ target orientations, and averaged across the two contexts (averaging is possible because of the symmetry of the $+20°$ and $-20°$ context orientations). $\theta$ is the step size between adjacent target orientations ($\theta = 1°$, $2°$, or $3°$, see the Methods; for the TI experiments having both $\theta = 1°$ and $\theta = 3°$ versions, there was no qualitative difference in $d'/\Delta\theta$ between $1°$ and $3°$, permitting an average over observers from both experimental versions). Results showed that sensitivity is mostly stable across RTs, possibly exhibiting a small improvement at the fastest RTs, and a small deterioration at the slowest RTs, clearly unlike the bias dynamics (Fig. 4, Supplementary Fig. 5). We note that due to the strong decision bias at shorter times and lack of sufficient trials, the $d'$ at shorter times may be under-estimated (see Supplementary Fig. 4). For individual data see Supplementary Fig. 7. Error bars are ± 1SEM.

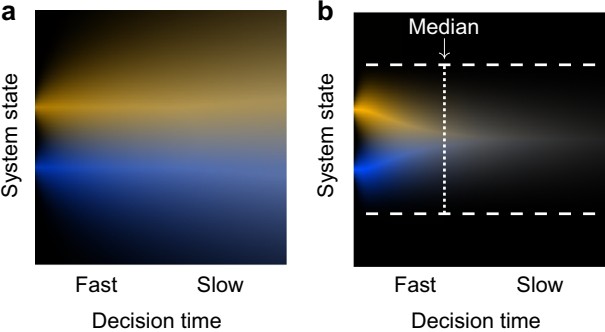

**Fig. 6 Bias reduction due to accumulated noise.** Probability density function (pdf) of the system state at different times, for different initial conditions (i.e., priors; top yellow vs. bottom blue), in **a** a diffusion process with no bounds, and **b** a diffusion process that stops when a bound (dashed lines) is reached. The effect of the initial state is lost with time (white color mix).

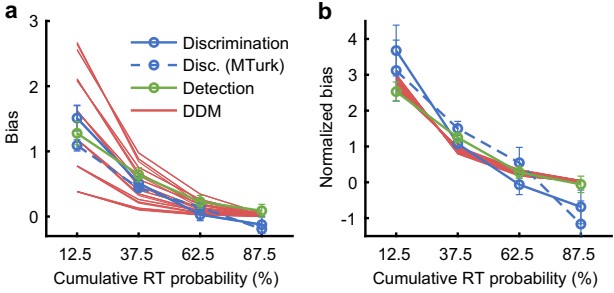

**Fig. 7 Reduction in prior-dependent bias is explained by the drift diffusion model. a** Bias (Eq. (1)) as a function of cumulative RT probability (i.e., quantiles), for behavioral data of the prior-dependent experiment (averages of Fig. 3a, d, g, individual data at Supplementary Fig. 2), and for a change in starting point in the bounded model (DDM) measured in four RT bins (red lines; the different groups of superimposed lines reflect different sizes of starting point changes away from the mid-point, and superimposed lines reflect varying drift rates; the influence of the bound-separation parameter is ignored because, for reasonable parameter values, it is mostly accounted for by changes in drift rate and starting point, by virtue of considering cumulative RT probability). **b** Bias from panel (**a**) divided by the average value (calculated separately for each observer and then averaged across observes). It can be seen that the stereotypical rate of reduction of bias owing to a starting point change in the DDM is qualitatively similar to behavior. Error bars are ± 1SEM.

Interestingly, in many stochastic processes (notably "memoryless" ones, such as random walks, Markov chains, and typical diffusion models), with processing time, the process gradually becomes independent of the initial state due to accumulated stochasticity (noise), so biases that reflect initial conditions are expected to gradually decrease with time. For example, in simple unbounded diffusion (random walk), the initial state is lost at a slow rate proportional to $\sqrt{\text{time}}$ (Fig. 6a), as in many stochastic processes. From the Bayesian perspective, the initial state measures the a priori information, and this prior is outweighed when more noisy evidence has been accumulated, leading to reduced prior effect in slower decisions.

Importantly, decision bias is reduced rapidly (~exponentially) in bounded decision models such as the DDM[10], in which the process of evidence accumulation continues until a bound is reached (Fig. 6b). For a starting point that is not too extreme (i.e., a moderate bias), the model predicts almost zero bias after the median decision time (Fig. 7a). Remarkably, the rapid rate of

reduction of bias was almost identical in the model and in the prior-dependent experiments (Fig. 7b). Indeed, a change of starting point in DDM provided an excellent account of behavioral prior-dependent bias, as indicated by AICc and BIC measures of the model fits (Table 1; differences relative to a "Fixed bias" model showed, Discrimination: $\Delta$AICc = $\Delta$BIC = −61, Discrimination MTurk: $\Delta$AICc = $\Delta$BIC = −204, Detection: $\Delta$AICc = $\Delta$BIC = −49; all $P < 0.001$ corrected, obtained by a random permutation test, and Bonferroni corrected for six multiple comparisons, see Methods). This result is consistent with earlier modeling of experimental data with the DDM[12,20,21]. Note that the theory presented here to explain decision time-dependent bias can also account for experimental situations where the bias effect persists with time (e.g., a change in the drift rate in the DDM), as evident with experimental manipulations affecting external noise selection[22]. More generally, persistent evidence selection should clearly lead to persistent bias effects[20,23].

With context-dependent bias, the observed rate of reduction in bias was slow (Fig. 4c, d). As such, neither a change of drift rate in the DDM (which predicts time-independent bias, Fig. 8a), nor a change of starting point in the DDM (which predicts a rapid reduction in bias, Fig. 8b), could by themselves explain the context-dependent bias (Fig. 8a, b, Table 1). We consider two alternatives. First, that bias is partially persistent, which is addressed within the DDM framework as a context-dependent change of both the starting point and the drift rate (i.e., a change of both the initial conditions and the rate of evidence accumulation, Fig. 8c). Second, that the process of evidence accumulation is unbounded rather than bounded (i.e., that decision timing is independent of the system state, as illustrated in Fig. 6a), leading to a $\sqrt{\text{time}}$ rate of reduction in bias (Fig. 8d). Qualitatively, both accounts offered an explanation for the slow rate of reduction in context-dependent bias (see Fig. 8c, d). Quantitatively, the unbounded alternative showed predictive performance that was occasionally much better than the bounded alternative (Table 1). Comparing the unbounded alternative to the "Fixed bias" reference showed a significant difference in predictive performance (Table 1, differences showing for all experiments $\Delta$AICc and $\Delta$BICc < −19, and $P < 0.001$ corrected). We note that DDM is sufficiently rich in parameters and may allow for more elaborated models[9]. A seemingly promising alternative to the accounts examined here is that the influence of the context is time independent but varies across trials[24], which can be modeled in the DDM as an intertrial variability in the drift rate[25]. However, this would predict the absence of line intersections in plots of report probability vs. time (Supplementary Fig. 9), unlike behavioral data (Fig. 2 and Supplementary Fig. 4). We note that the sign of the context-dependent bias is opposite to the prior ("anti-Bayesian"), consistent with a shift of the reference (i.e., bound positions) in the direction of the prior.

Our main goal here is to explain the dependency of decision bias on RT. The theories considered may also be tested for other aspects of the data, such as RT distribution and dependence of discrimination sensitivity ($d'$) on RT. The latter is found here to be roughly independent of RT, as predicted by the standard DDM[26], and by design by its predecessor sequential probability ratio test (SPRT)[27]. To model RT distributions, more trials are required. We note that the factor we use for analysis, RT, is an observable rather than an experimentally manipulated factor. This design decision raised the need to control for potential confounding factors (as discussed in the text), but importantly, note that an observable cannot always be replaced by an experimentally manipulated factor. Notably, under the assumptions of the DDM, the stochasticity in diffusion underlies the co-variability of RT and bias. Therefore, considering the

**Table 1 Descriptive model comparison.**

| | Experiment | Comparison method | Model | | | |
|---|---|---|---|---|---|---|
| | | | Fixed bias (DDM drift rate change) | DDM start point change | DDM, start point and drift rate change | Unbounded, start point change |
| Prior-dependent | Discrimination | AICc | 82 | **20** | 24 | 59 |
| | | BIC | 85 | **23** | 26 | 61 |
| | Discrimination MTurk | AICc | 504 | **301** | 303 | 417 |
| | | BIC | 637 | **432** | 437 | 551 |
| | Detection | AICc | 83 | **35** | 24 | 54 |
| | | BIC | 90 | **42** | 31 | 61 |
| Context-dependent | TI until-response | AICc | 131 | 116 | **77** | **64** |
| | | BIC | 140 | 125 | **86** | **72** |
| | TI 200 ms | AICc | 124 | 125 | 128 | **84** |
| | | BIC | 133 | 134 | 137 | **92** |
| | TI MTurk until-response | AICc | 580 | 846 | **495** | **489** |
| | | BIC | 742 | 1008 | **659** | **653** |
| | TI MTurk 200 ms | AICc | 589 | 793 | 562 | **519** |
| | | BIC | 732 | 936 | 707 | **664** |
| | TI MTurk mix | AICc | 5083 | 6928 | 5006 | **4673** |
| | | BIC | 7406 | 9250 | 7333 | **7000** |
| | TAE fixation | AICc | 154 | 158 | **118** | **117** |
| | | BIC | 168 | 172 | **132** | **130** |
| | TAE fixation Pinchuk et al.[16] | AICc | 159 | 104 | **76** | 87 |
| | | BIC | 170 | 115 | **87** | 98 |
| | TAE periphery | AICc | 160 | 166 | **132** | 134 |
| | | BIC | 178 | 185 | **150** | 152 |
| | TAE periphery non-retinotopic | AICc | 148 | **122** | 116 | 128 |
| | | BIC | 166 | **140** | 134 | 146 |

Shown are Akaike information criterion (AICc) and Bayesian information criterion (BIC) values which indicate how well a given model accounts for the measured behavioral bias compared to other models, when taking into account the number of fitted model parameters. Within a row, lower values indicate better model performance, where the standard rule-of-thumb is that a difference of at least ten can be interpreted as a "very strong" evidence in favor for the winning model[42]. Correspondingly, models having a difference of approximately ten from the winning model are in bold typeface. (Between rows, differences mostly reflect differences in the amount of data.) In all models, a single parameter was fit per observer, corresponding to the magnitude of the individual bias. In addition, a single group-level parameter was fit in two of the models: the "DDM, start point and drift rate change" model with the group level parameter indicating the percentage of the overall bias that is due to a starting point rather than a drift rate change; and the unbounded model where the group level parameter was the nondecision time, namely, $t_0$. Results show, for both prior-dependent and context-dependent bias, the validity of a model which assumes a change in the starting point of the process, as indicated by reduced AICc and BIC values relative to the "Fixed bias" null-hypothesis model.

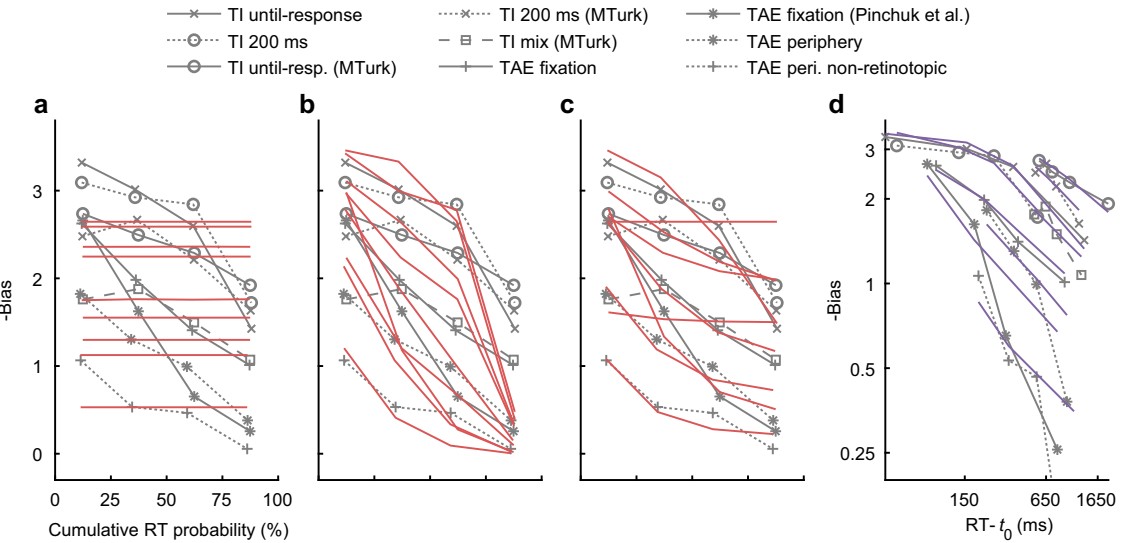

**Fig. 8 Modeling the reduction in context-dependent bias.** Shown is bias (Eq. (1)) as a function of RT, averaged across observers, in models and in the behavioral context experiments (gray lines, reproduced from Fig. 4 cd). **a–c** Bounded model (DDM, red lines), measured in four RT bins, where the influence of a context change (−20° vs. +20°) is modeled as a (**a**) drift rate change, **b** starting point change (same model as in Fig. 7), **c** drift rate and starting point change (see the Methods). **d** Unbounded model (purple lines), where the influence of a context change is modeled as a change in starting point in an unbounded diffusion process. This model was analyzed by fitting bias to a $\sqrt{RT - t_0}$ decay rate function (see the Methods). The $t_0$ measures the non-decision time (10–550 ms; using a fixed value of 350 ms obtained from the RT distributions gives similar results). Note that the x-axis in panels **a–c** is for relative RTs (i.e., cumulative RT probability), while panel **d** is for physical RT (time difference from $t_0$), which is reasonable considering the different models. Fitting was done on the group level. Overall, the different modeling approaches (panels **c**, **d**) can qualitatively account for a slow rate of reduction in bias.

slower/faster subset of trials (as we do here) is not immediately interchangeable with experimental manipulations that affect the average RT by changing the separation between the bounds. Overall, we find an excellent account for both prior-dependent and context-dependent biases by assuming a change in the initial conditions of evidence accumulation (with or without bounds, and with or without a change in the rate of evidence accumulation).

## Discussion

Overall, this work illustrates an innovative way of thinking about bias and time in human decision-making. Instead of fitting a specific model to data (as in ref. [20]), we make a simple yet powerful general claim: that bias derived from starting conditions (e.g., prior) gradually decreases with decision time. This claim applies to an entire family of stochastic decision processes, which can be indistinguishable with limited data, emphasizing the importance of focusing on the general principle.

Most importantly, the finding of reduced bias with longer decision times may appear to perfectly conform with a dual-theory account of a transition between separate systems[7,28,29]: from a fast system that is bias-prone, to a slow system that is bias-free. Similarly, the known reduction in context-dependent bias with the duration of testing stimulus presentations is currently explained by assuming dual processing (in low vs. high systems)[5,6,18]. However, we have found that using models based on the decision principles described above offers a full mechanistic explanation. Even a rapid reduction in bias, with the slower half of responses measuring practically zero bias, can be explained by bounded evidence accumulation (that is, a rule for when to stop accumulating evidence that depends on the extent of evidence that has been accumulated). This account has very strong support in brain decision making (DDM[8,9,11]) and clear statistical implications (SPRT[12,13,27]). Of course, there can be dual processes in the brain, and there are low vs. high brain areas, but at least for the basic perceptual decisions considered here, there is a simple mechanistic account that does not need to assume multiple systems. Note that it is still possible that the different model parameters are set by two or more brain or cognitive modules. For example, the starting point (prior) is possibly set by a higher-level experience-based module, which integrates information over extended time scales, while the information accumulation is possibly performed by a lower-level module that closely follows the sensory input. The specific anatomical analogue may depend on the behavioral task at hand.

Traditionally, perceptual phenomena are considered to be a part of the faster system[28], although there is no clear definition. One attempt at formulating a definition by Evans and Stanovich[29] suggests that "rapid autonomous processes (Type 1) are assumed to yield default responses unless intervened on by distinctive higher order reasoning processes (Type 2)", which at least phenomenologically appear to match the reduction in bias discussed here. Regardless of terminology, perceptual decisions and the simple mechanistic models they permit should be considered in the discussion about fast vs. slow systems.

## Methods

**Observers**. The purpose of this study was to investigate the interaction between bias and decision time for prior- and context-dependent biases. The experiments reported here were performed in a laboratory or with a web-based interface through the Amazon Mechanical Turk (MTurk). The work was carried out in accordance with the Code of Ethics of the World Medical Association (Declaration of Helsinki), and was approved by the Institutional Review Board (IRB) of the Weizmann Institute of Science. All observers were naïve to the purpose of the experiments.

In the laboratory experiments, $N = 43$ observers participated (32 females, 11 males, aged $26 \pm 4$, Mean $\pm$ SD, median of 25, in the range 18–40), including data from eleven observers obtained with permission from ref. [16]. One additional observer dropped out after the first session (for personal reasons), and her data were not analyzed. Most observers were students of the Rehovot Campus of the Hebrew University of Jerusalem (i.e., Faculty of Agriculture, Food and Environment). Observers were recruited by advertisement, and began participation in the experiments conditioned on passing an eye examination. Observers were compensated at a rate of 50 NIS per daily session. Observers had normal or corrected-to-normal vision, and have provided their written informed consent.

In the MTurk experiments, $N = 636$ observers participated (reported age of $36 \pm 11$, median of 33, in the range 18–74; gender information was not always collected). Additional $N = 228$ observers were excluded, as described below. Observers were recruited through the MTurk platform, with an offered compensation that was calculated based on an approximate $7.5/h rate. The typical selection criteria were as follows: "PercentAssignmentsApproved" ≥99, "NumberHITsApproved" ≥2000, "LocaleCountry" = "US". By the nature of the platform, participants may stop performing the experiment at will, without providing a justification. Of the participants who "Accepted" the task, roughly 37% completed the offered experiment and received compensation. The web-based study was reviewed by the Institutional Review Board (IRB) of the Weizmann Institute of Science and deemed exempt from the collection of informed consent forms.

**Apparatus**. Laboratory setup: The stimuli were presented on a 22″ HP p1230 monitor operating at 85 Hz with a resolution of $1600 \times 1200$ that was gamma-corrected (linearized). The mean luminance of the display was 26 cd m$^{-2}$ (detection, discrimination, and TAE experiments) or 49 cd m$^{-2}$ (TI experiments) in an otherwise dark environment. The monitor was viewed at a distance of 100 cm.

Amazon Mechanical Turk setup: To display time-accurate full-screen stimuli in the web-browser of arbitrary MTurk observers, we developed a dedicated JavaScript client for the PC versions of the Chrome, Mozilla, and Opera web browsers. Using the web-browser version of the OpenGL technology, namely, WebGL, permitted a highly efficient implementation whereby the client machines typically achieved single-frame time accuracy. Clients were required to have a GPU-enabled WebGL implementation, as indicated by the "failIfMajorPerformanceCaveat" API flag. An additional performance test was conducted, to ensure that the client machine is able to achieve the required time accuracy. Importantly, during an experiment, estimations of the stimuli presentations timing were collected. Post-hoc timing diagnostics showed that most trials, of most observers, measured nearly perfect single-frame accuracy. For example, in the TI MTurk 200 ms experiment, the 95% worst-case timing error relative to the within-observer mean was 24 ms. For the "Discrimination MTurk" experiment, the 95% worst-case error was 16 ms. The display luminance, gamma calibration, sitting distance, and environmental lightning conditions were only controlled to the degree reflected in the task performance (see below). We note that all statistical comparisons we report are within-observer (paired), hence reflect measurements from the same display. Also, note that the absence of a gamma-correction precludes performing experiments that require pure sine-wave frequencies (e.g., measuring the frequency-selectivity of an effect). RTs were measured locally on the client machine, so that internet connection speed did not affect the measured RT. As seen in Supplementary Tables 1 and 2, the reliability of the crowdsourcing setup was quite high. For example, 48 out of 50 observers measured an RT-dependent reduction in prior-dependent bias in the "Discrimination MTurk" experiment.

**Stimuli and tasks**. All stimuli were presented using dedicated software on a uniform gray background. To begin stimulus presentation in a trial, observers fixated on the center of the display and pressed the spacebar (self-initiated trials). Responses were provided using the left and right arrow keys. As described below, the used Gabor patches were of the same parametrization (except orientation, monitor position, and contrast).

In MTurk, the stimuli, defined in pixels, were resized using linear interpolation to maintain a fixed proportion relative to the full-screen monitor resolution. To minimize resizing artifacts, the resizing multiplier was rounded down to the nearest lower multiple of 0.25 (e.g., resizing by a multiplier of 1.25 instead of 1.33).

Discrimination (2AFC) experiment (Lab): Stimuli were Gabor patch targets tilted $-0.5°$ or $+0.5°$ relative to vertical (50 ms presentation, 50% Michelson contrast[30], Gaussian envelope $\sigma = 0.42°$, spatial frequency wavelength $\lambda = 0.3°$, random phase, 750 ms onset ±up to 100 ms onset jitter). Observers were instructed to report whether the orientation of the target is CW or CCW to vertical (2AFC), with auditory feedback indicating mistaken reports. Four peripheral crosses co-appeared with the target.

Discrimination (2AFC) experiment (MTurk): Stimuli were Gabor patch targets tilted $-4.5°$, $-1.5°$, $+1.5°$, or $+4.5°$ relative to vertical (~66 ms presentation, 300 ms onset ±up to 100 ms onset jitter). Gabor targets had an amplitude of 64 gray levels (out of 256, i.e., corresponding to a contrast of 50% in a linearized display), $\sigma = \sqrt{2}\lambda$, $\lambda = $ ~2% of screen height, and a random phase. Observers were instructed to report whether the orientation of the target is CW or CCW to vertical (2AFC), with visual feedback indicating mistaken reports (a large "X" presentation).

Detection experiment: In the target-present trials, the stimuli were low-contrast Gabor patches (50 ms presentation, 0.5–1% contrast, vertical orientation, $\sigma = 0.42°$, $\lambda = 0.3°$, random phase, 750 ms onset ±up to 100 ms onset jitter). Observers were instructed to report whether the target appeared or not, with auditory feedback indicating mistaken reports. Four peripheral crosses co-appeared with the target presentation interval (in both target-present and target-absent trials).

TAE experiments: The following presentation sequence was used (Fig. 4b): a blank screen (600 ms presentation), Gabor "adapter" (i.e., context, oriented −20° or +20° to vertical, 50 ms), a blank screen (600 ms), and a near-vertical Gabor "target" (50 ms). Observers were instructed to inspect the adapter and target presentations, and then to report whether the orientation of the target was CW or CCW to vertical (2AFC, no feedback). Gabor patches were 50% Michelson contrast with $\sigma = 0.42°$, $\lambda = 0.3°$, and random phase. Two versions of the experiment were run: fixation and periphery. In the fixation experiment, adapters and targets were presented at the fixated center of the display, and targets were oriented −9° to +9° (in steps of 1°). In the periphery experiment, adapters and targets were presented at either left or right of the fixation (at ±1.8° eccentricity). The target was presented either at the same side as the adapter (retinotopic) or at the opposite side (non-retinotopic), randomly. Targets were oriented from −12° to +12° (in steps of 2°). The reason that the TAE experiments had no variability in the onset of the target is that the time difference between adapter and target is known to affect the magnitude of the TAE[31]. Four peripheral crosses co-appeared with the target to facilitate discrimination between the adapter and the target.

TI experiments (Lab): Stimuli (e.g., Fig. 4a right) consisted of a near-vertical sine-wave circular "target" (oriented from −9° to +9° in steps of 1°, $\lambda = 0.3°$, random phase, a radius of 0.6°), and a sine-wave "surround" annulus (oriented −20° or +20°, $\lambda = 0.3°$, random phase, width of 1.2°, and a gap of 0.15° from the central circle). The sine-wave gratings had a contrast of 100%. Observers were instructed to inspect the target, and to report its orientation as CW or CCW to vertical (two-alternative forced choice, 2AFC; no feedback). The target + surround stimuli were presented starting from 450 ms after the trial initiation (±up to 100 ms jitter), for a duration of either 200 ms ("200 ms" experiment) or for however long it took the observer to provide a response ("until-response" experiment).

TI experiments (MTurk): The main stimuli used were nearly identical to the lab, resized by using a conversion of ~5.5% of screen height for 1° of visual angle in the lab stimuli. The sine wave gratings had an amplitude of 128 gray levels (corresponding to a contrast of 100% in a linearized display). The target orientation steps were either 1° (as in the lab) or 3°, and the onset jitter was not always used (in which case the onset was fixed at 350 ms after the trial initiation).

**Procedure**. In the laboratory, most observers participated in multiple experiments of this study, separated by at least 3 days of a break. We required a very low rate of finger errors (less than one reported mistake in 200 trials), to minimize contamination by fast guesses[32]. All observers managed to achieve this level of accuracy with persistent coaxing. Observers were encouraged to respond quickly, unless the speed of response led to mistakes. Under all experiments, each daily session was preceded by a brief practice block with easy stimuli (this practice was repeated until close-to-perfect accuracy was achieved).

In the MTurk, observers participated in a single experiment, for a single daily session, as indicated by having a unique "Worker ID". To participate in an experiment, MTurk observers had to achieve a near-perfect accuracy in a brief practice block with easy stimuli. Finger errors were partially controlled for by pruning low performing MTurk observers. Specifically, MTurk observers were excluded based on the parameter thresholds described in Supplementary Table 3. The thresholds were intended to be relatively lenient, so as to not exclude nearly any of the cooperative observers who performed the task(s) as instructed. The exclusion criteria were predetermined for all experiments, with the exception of the "TI mix" dataset (see Supplementary Method 1).

Discrimination (2AFC) experiment (Lab): Observers ($N = 7$) completed a single session. Trials were blocked into 80 trials with a given prior (either 75 vs. 25, 50 vs. 50, or 25 vs. 75% for the −0.5° vs. +0.5° orientation alternatives). In each session, a set of six blocks (two per prior, in random order, each lasting ~2 min, separated by a 20-s break to minimize inter-block contamination by learned priors) was completed twice (totaling four blocks per prior; ~13.5 min per set, separated by a 2-minute break). Observers were informed that different blocks may have different priors of the decision alternatives. Two observers were disqualified before data analysis, one having anomalously high sensitivity ($d' = 2.5$, with other observers showing $d' = 1.01 \pm 0.18$, mean ± SD), and one having an anomalously strong baseline response bias in favor of one of the alternatives, which saturates the measured probabilities. (Both disqualified observers exhibited a prior-dependent bias in fast replies and no prior-dependent bias in slow replies.) Note that the experimental design assumes that all observers have comparable sensitivity. The horizontal orientation of the monitor and the table was verified using a spirit level.

Discrimination (2AFC) experiment (MTurk): Observers ($N = 50$; additional $N = 34$ observers were excluded based on the predetermined criteria described in Supplementary Table 3) completed a single session. Trials were blocked into 80

trials with a given prior, using priors of 75 vs. 25, and 25 vs. 75%, for the CCW vs. CW orientation alternatives (of both 1.5° and 4.5° target orientations). Observers completed a total of 10 blocks (5 per prior alternative), separated by breaks of at least 10 s, lasting ~30 min (sometimes more). Observers were not informed that different blocks have different priors. Compared with the laboratory design, the MTurk version was intended to be less rigid.

Detection experiment: Observers ($N = 9$) completed from two to three daily sessions. Trials were blocked into 80 trials with a given prior (either 75 vs. 25, 50 vs. 50, or 25 vs. 75% for target-present vs. target-absent). In each session, two sets of six blocks were completed (two per prior, in random order). Blocks (lasting ~2 min) were separated by a 15-s break. The two sets (~14.5 min each) were separated by a 2-minute break. To ensure a stable $d'$, the level of difficulty was adjusted per observer at the beginning of each session with a staircase procedure, and occasionally also during the break between the two sets.

TAE experiments: Sessions were composed of blocks of 125 trials (lasting ~5 min), separated by 2-min breaks of blank screen free viewing. In the fixation experiment, observers ($N = 12$) performed a single session consisting of six blocks. In the periphery experiment, observers ($N = 14$) performed 3–8 daily sessions each consisting of five blocks.

TI experiments (Lab): Sessions consisted of blocks of 190 trials (lasting ~5 min), separated by 2-min breaks of blank screen free viewing. Observers ($N = 10$ and $N = 10$, in "200 ms" and "until-response" experiments, respectively) performed a single session consisting of five blocks.

TI experiments (MTurk): In the "200 ms" and the "until-response" experiments, observers ($N = 53$ and $N = 58$, respectively; additional $N = 17$ and $N = 5$, respectively, were excluded) performed a single session consisting of four or five blocks of 190 or 196 trials. Some of the supplementary analyses depended on the step size of the target orientation range (either 1° or 3°, see above). Blocks lasted ~5 min and were separated by breaks of at least 2 min (sometimes more). The exclusion criteria were predetermined.

**Additional data**. TAE fixation experiment from Pinchuk et al.: We used, with permission, data from the "No expectation" and "Control" conditions reported in[16], which used a nearly identical stimuli and procedure as the TAE fixation described above ($N = 11$ observers).

TI "mix" dataset: We used data pooled from a number of TI experiments. The stimuli and task were similar to the ones used here (Fig. 4a), with modifications described in Supplementary Method 1. Data were collected using the Amazon Mechanical Turk from $N = 475$ observers (no overlap with the other conditions reported in this work).

**Analysis**. Trial pruning: For all data (laboratory and MTurk), single trials were pruned based on RT. In all context experiments (TAE and TI), trials with RT < 300 ms were pruned (this rule was intended to address the rare occasions where the observers judged the perceived orientation of the adapter instead of the target in the TAE experiments). In all MTurk experiments (prior-dependent, TAE, and TI), trials with RT slower than 10 s were pruned.

Data binning: To measure the interaction of time and bias, behavioral data were binned, based on RT into $N$ bins. For the detection and discrimination experiments, binning was carried out separately for each combination of trial stimulus type × experimental block (note that different blocks correspond to different experimental priors). For example, separately for target-present and target-absent trials of each block. For the TAE and TI experiments, binning was carried out separately for each combination of experimental day × target orientation × context orientation (adapter or surround). Under the TAE periphery experiments (retinotopic and non-retinotopic), binning combinations were further conditioned based on adapter side × target side. When the number of trials was not an exact multiple of the required number of bins, a deterministic rule was used. Binned trials were pooled from all relevant repetitions, separately for each observer. Conceptually, the strict binning rule we applied prevents/minimizes confounds, and in particular, prevents interaction between trial type and the bin used (because different target stimuli, under different priors/contexts, can have different inherent difficulties and hence, different average RT[11]). As a sanity check, to verify that the results are not contingent on the chosen analysis, we also tried less strict binning combinations. For example, ignoring the prior/context in binning leads to perfect alignment of the time bins in Fig. 2 and Supplementary Fig. 4. We found that any reasonable binning rule we tried that had more trials than binning combinations led to nearly identical results. In the prior-dependent experiments, the first ten trials were excluded from the analysis, so that the priors could be learned (which takes at least a few trials to learn, e.g.[33]).

Bias and sensitivity calculation: To quantify bias and sensitivity, we relied on measures motivated by SDT[1]. Specifically, we used bias (Eq. (1)) and $d'$ (Eq. (3)). The definition of bias (Eq. (1)) is equivalent to the average change in the internal criterion $c$ (Eq. (2)) (as seen in Fig. 3). Formally, define $p_d^s$ to be the response probability (the percent of "+" responses) for stimulus $s$ ($s = $ "−" or $s = $ "+") in condition $q$ (taking one of two values, for two prior probabilities, or two context orientations). The average change in bias of the two stimuli is then $0.5 \cdot (\text{Bias}^+ + \text{Bias}^-)$, and the shift of the internal criterion is $c_2 - c_1$ (where $c_q$ is

the internal criterion in condition $q$). Thus

$$
\begin{aligned}
0.5 \cdot (\text{Bias}^+ + \text{Bias}^-) \\
= 0.5 \cdot \left( \left( z(p_1^+) - z(p_2^+) \right) + \left( z(p_1^-) - z(p_2^-) \right) \right) \\
= -0.5 \cdot \left( z(p_2^+) - z(p_2^-) \right) + 0.5 \cdot \left( z(p_1^+) - z(p_1^-) \right) = c_2 - c_1.
\end{aligned}
\tag{4}
$$

Therefore, the average change in bias of the two stimuli is equal to the shift of the internal criterion.

To avoid saturation, probabilities were clipped to the range $\left[ \frac{1}{2n}, 1 - \frac{1}{2n} \right]$, where $n$ is the number of trials in a measurement. Bias and $d'$ were calculated by applying Eqs. (1) and (3), respectively, after the probabilities were clipped to the above range. Probabilities were calculated after pooling RT-binned trials from all relevant days, blocks, spatial positions, and target stimuli (see below).

In the prior-dependent experiments, bias was calculated separately for each stimulus (as in Fig. 3, Supplementary Fig. 1), then averaged (Fig. 7, Supplementary Fig. 2). In the "Discrimination MTurk" experiment, only the ±1.5° target stimulus orientations were analyzed (the ±4.5° orientations were intended to stabilize the observers' performance).

In the context-dependent experiments, the range of target orientation was continuous, having steps of 1°, 2°, or 3°. (The use of a continuous range permits measuring the bias in degrees, see Supplementary Fig. 8). In the experiments with 1° steps, the $d'$ between adjacent orientations was small ($d' \approx 0.35$, see Fig. 5). In these experiments, we pooled data from a pair of adjacent target orientations when calculating the bias. We chose, separately for each observer, the pair of orientations having discrimination performance that is closest to chance level. This approach improves the dynamic range over calculating bias for a single, fixed target orientation, especially under less-controlled conditions (MTurk) (data for single, physical orientations is reported at Supplementary Figs. 4 and 5). In the experiments with 2° and 3° steps, the $d'$ between adjacent orientations was ~1 (Fig. 5). For these experiments, bias was calculated for the single orientation that is closest to chance level. It can be observed that the measured bias for most observers was not saturated (Supplementary Figs. 2 and 6). An alternative approach for avoiding saturation is to average probabilities across observers before calculating bias and sensitivity. This approach led to the same qualitative findings as reported here. Moreover, the same results were found when measuring bias as the shift in the PV orientation (described below; Supplementary Fig. 8).

Fitting perceived orientation: In addition to the bias measure motivated by SDT (described above), we also measured context-dependent bias using the magnitude of the shift in degrees in the perceived target orientation (reported in Supplementary Fig. 8). Specifically, the magnitude was defined as half the shift in the PV orientation between the two context orientations. To find the PV orientation in a given condition, the percentage of CW reports as a function of the target orientation was interpolated to find the orientation with 50% CW reports (i.e., equal probability for CW and CCW reports; fitting to a cumulative normal distribution that takes into account the lapse rates was achieved with the Psignifit 3.0 software[34]). Under the TAE periphery conditions (retinotopic and non-retinotopic), the data reflected two "target" sides measured at several experimental days. The effect magnitude was calculated separately for the different experimental days and sides, then averaged across days and sides. Note that although this method of analysis is standard for measuring the TAE and the TI, it is less meaningful when binning RT, because different target orientations have different difficulties, and hence, different mean decision times. Specifically, if binning is done based on time irrespective of target orientation, then the bins are unbalanced (e.g., the fastest bin will only contain trials of easy target orientations). If binning is done separately for each target orientation (as we do here), then the bins are balanced, but there is no descriptive time range associated with a bin (i.e., bins reflect relative rather than physical times).

**Model comparison**. We wish to stress that our main point is not related to a specific model, but to a general theoretical idea that can be implemented using different models. For the purpose of comparing how well different mechanistic models can account for the observed time-dependence of bias, we followed recommendations[35,36] and considered both the generative performance (how well the model can generate the data, Figs. 4 and 8), and the predictive performance (how well the model can predict the data, Table 1). Both generative and predictive performances were obtained by fitting the model to the data (there was no need for simulations, because the models under consideration have analytical expressions). The illustrations in Fig. 6 were obtained by simulations (100,000 trials).

Unbounded model: The model assumes some diffusion process of evidence accumulation, equivalent to a simple random walk (a Wiener process, Fig. 6a). The process starts from some initial condition, and gradually diverges due to stochastic diffusion (noise). In this case, we expect the influence of a change in the starting point of the random walk to diminish at a rate proportional to a square-root of the time, that is

$$
\text{Bias} = \frac{b_0}{\sqrt{t - t_0}},
\tag{5}
$$

where $t$ is time, $t_0$ is the initial time (i.e., the non-decision time), and $b_0$ is the initial bias at time $t_0$. Note that the unbounded model does not explain the decision time itself, just how the decision time affects the bias. The $t_0$ was restricted to the range 10–1000 ms.

When fitting to the unbounded model, the predictor was physical time (average RT in a bin), and the predicted bias was given by Eq. (5) where $t$ is the predictor, and $b_0$ and $t_0$ are the two free parameters. Predictions were clipped to the approximate range used for the behavioral measurements (see above). The $b_0$ parameter was fit per observer, while the $t_0$ parameter was fit on the group level (separately for each experiment). Because $t_0$ represents the non-decision time, we also verified that using a fixed value of $t_0$ (350 ms), obtained from the behavioral RT distributions, leads to the same findings as found when $t_0$ is fitted from the bias data. Note that under this model the predictor is physical time, so different observers can have different predictor values.

Bounded model (DDM): An alternative approach of modeling decision processes, which also explains the decision times, is to assume that there are decision bounds. When the accumulated evidence reaches a bound, the process is stopped and a decision is made. Here, we consider the standard bounded model (DDM[8–11,37], Fig. 6b; see mathematical background in refs. [38,39]). The DDM can be defined using four parameters: the drift rate ($v$), bound separation ($a$), starting point ($z$), and non-decision time ($t_0$). In this description, the bounds are at 0 and $a$, and the process starts from point $z$. Under these conditions, the analytical expression of the probability distribution of the decision times is known, and for the lower bound is given by[40]

$$
f(t|v, a, z) = \frac{\pi}{a^2} \exp\left( -\frac{vaz}{a} - \frac{v^2 t}{2} \right) \sum_{k=1}^{\infty} k \exp\left( -\frac{k^2 \pi^2 t}{2a^2} \right) \sin\left( \frac{k\pi z}{a} \right).
\tag{6}
$$

The distribution of decision times for the upper bound is similar, given by setting $v' = -v$ and $z' = a - z$. Additional parameters can be introduced to the DDM, such as intertrial variability parameters[9]. Here, we did not consider such extensions, with the exception of Supplementary Fig. 9b, where the intertrial variability of the drift rate was used[25] (computed using the fast-dm-30 software[41]).

When fitting to the DDM, the predictor was relative time (cumulative RT probability, i.e., quantiles), and the predicted bias was obtained by changing the drift rate and/or the starting point. We considered three modeling approaches. (1) "Fixed bias" model (one parameter per observer). The model is given by assuming that the bias is fixed in time, or equivalently, that the bias is a consequence of a change in the drift rate of the DDM (i.e., a change of $v$ from + to −, the single parameter being the size of this change). (2) "Start point change" model (one parameter per observer). The model is given by assuming that the bias is caused by a change in the starting point (i.e., a change of $z$ from $\frac{a}{2} + b$ to $\frac{a}{2} - b$ for some $b$, the single parameter being the size of this change). (3) "Start point and drift rate change" model (one parameter per observer, one parameter on the group level). The model is given by assuming that the bias is caused by a change in both the drift rate and the starting point. The two parameters used were the size of the overall change, $b$, and the proportion of the change that is applied to the starting point over the drift rate, $p$. Then, bias is given by a change in both starting point ($z$, changed from $\frac{a}{2} + b \cdot p$ to $\frac{a}{2} - b \cdot p$) and drift rate ($v$, changed from $+b \cdot (1 - p)$ to $-b \cdot (1 - p)$). The $p$ parameter was fit on the group level (separately for each experiment).

The bias between two DDM parametrizations (that differ by starting point and/or drift rate) was obtained by computing the analytical distributions of the upper and lower bounds for each parametrization (Eq. (6)), binning by RT, clipping probabilities to the approximate range used for the behavioral measurements (see above), and measuring the bias (Eq. (1)). This results in bias as a function of cumulative RT probability. Note that this function is practically invariant, within the relevant range of parameters used, to the value of the drift rate, bound separation, and non-decision time parameters, if they are fixed (not changed between the two parametrizations). We verified this claim by using simulations, see examples for invariance at Fig. 7 and Supplementary Fig. 9a.

Fitting: Fitting of parameters was achieved by minimizing the squared difference between the predicted and the measured bias. Fitting was done on the individual or hierarchical level (i.e., always one parameter per observer, possibly with an additional parameter on the group level, as described above). Fitting was achieved by using the "fminsearch" function of MATLAB® R2019b.

AICc and BIC: To quantify the predictive performance, we used the small-sample correction of the Akaike information criterion (AICc), as well as the Bayesian information criterion (BIC). First, to calculate the likelihood of the data given a fitted model, samples were assumed to be drawn from a normal distribution around a model prediction (with standard deviation equaling to the standard deviation of all errors). This assumption permits calculating the log-likelihood (LL) of the data for the maximum-likelihood model by using the sum of the squared errors (SSE) of the least-squares model, according to: $\text{LL} = -\frac{1}{2} n \left( \ln\left( 2\pi \frac{\text{SSE}}{n} \right) + 1 \right)$, where $n$ is the number of samples and $\ln(\cdot)$ is the natural logarithm. Then, $\text{AIC} = -2\text{LL} + 2K$, and $\text{BIC} = -2\text{LL} + \ln(n)K$, where $K$ is the number of fitted model parameters plus one (for the fitted "parameter" of the standard deviation of the errors). Finally, $\text{AICc} = \text{AIC} + \frac{2K(K+1)}{n-K-1}$, which is the recommended approximation of the small-sample correction for AIC[42]. The small-sample correction was used because $n/K < 40$.

**Statistics and reproducibility**. Linear mixed-effects regression: The statistical test used to determine if measured bias depends on time was a linear mixed-effects

regression analysis. The linear model had a fixed intercept term, a fixed slope term using the relative RT (i.e., time bin index, 1:4), and a random intercept term grouped by observer (similar to a repeated-measures ANOVA). This model can be described using Wilkinson notation as: "Bias ~ RT_ bin_index + (1 | Observer)". The use of relative rather than physical time is important because the distribution of RTs is skewed (as seen in all figures here that depict RTs, note that the scaling is always logarithmic, and as discussed elsewhere[25]). The use of linear mixed-effects regression rather than, e.g., repeated-measures ANOVA permits analyzing systematic time-dependent trends in the data. We used a Bonferroni correction for two multiple comparisons because we considered using either four or six time bins. Using six bins led to saturation of the bias measurements in the TI experiments, so we decided to use four bins. (The findings we report remain significant when using either four or six bins, with the slope term always showing $P \leq 1 \times 10^{-4}$ corrected for two multiple comparisons.) We note that other analysis decisions were motivated by best practice/necessity, so they did not inflate the alpha. The main idea we followed was to be as strict as possible, to prevent/minimize confounds.

Effect sizes: To compute Cohen's $d$ for the difference between the fastest and the slowest bins, we consider the mean difference divided by its standard deviation: Cohen's $d = \frac{\overline{Y_1} - \overline{Y_2}}{S_{Diff}}$, where $\overline{Y_1}$ is the mean bias in the fastest bin, $\overline{Y_2}$ is the mean bias in the slowest bin, and $S_{Diff}$ is the sample standard deviation of the pairwise differences[43,44]. This is the calculation employed by the standard $R$ statistical programming language in the paired-samples case (effsize v0.7.4)[45,46]. We also report Hedge's $g$, which is the small-sample correction to Cohen's $d$[43,47]: Hedge's $g = d\left(1 - \frac{3}{4n-1}\right)$, where $n$ is the number of samples (i.e., observers).

Random permutation tests: We consider the null hypothesis that bias is independent of RT. To test if a proposed model is significantly better than expected under the null hypothesis, we employed a random permutation test. Specifically, in $N = 12,000$ random repetitions, the RT of all trials was randomly permuted, and the ΔAICc was recorded for each iteration, where the ΔAICc is the difference in AICc between the proposed model and the "Fixed bias" null hypothesis model. The distribution of ΔAICc under the null hypothesis was then compared to the ΔAICc measured behaviorally (one-tailed, because only one side of the distribution indicates that the proposed model is better). Specifically, the probability to measure under the null hypothesis ΔAICc values that are at least as extreme as the behaviorally measured ΔAICc was recorded (with a minimum of $\frac{1}{N/2} = 0.006$). The same test was used for ΔBIC. We then applied a Bonferroni correction for six multiple comparisons, because there were three considered models, and two options for the number of time bins (see above). Therefore, the lowest possible $P$ value was $\frac{0.006}{6} = 0.001$. The reason we used a permutation test is that the standard likelihood ratio test (and similar alternatives) cannot be used for non-nested models. Additionally, the permutation test is nonparametric, supplementing the remaining statistical tests used here which are parametric.

Sample size: Here, we use standard psychophysical methods employed in vision research. The effects studied (response bias, TAE and TI) are expected to be found with each observer tested in standard experimental conditions, as indicated by previous work (response bias[1], TAE[6,16,48], and the TI[49]). The reader is invited to try (aftereffects[3,50], TI[51,52]). The behavior studied here concerns the dependency of these robust effects on RT. We estimated that, if a strong RT-dependence exists (as predicted by a change in starting point in the DDM), then the effect size should be very large (e.g., Cohen's $d > 1.5$). Although the context-dependent experiments showed shallower time-dependence than we initially predicted, the statistical analysis confirmed the robustness of the results (see Supplementary Tables 1 and 2). Our laboratory sample sizes were somewhat arbitrarily set ($7 \leq N \leq 14$), taking advantage of existing unpublished RT data collected in the lab (testing properties of the TAE and the TI; some of them from previously published experiments, e.g.[16]), and having at least seven participants per experiment (a standard number used in the field for experiments of this kind). To ensure that the findings are robust and easily replicable, we also report data collected using the Amazon Mechanical Turk ($N \geq 50$ per experiment). We estimated, using power analysis based on the laboratory study and our experience with the reliability of observers in the MTurk platform, that 10–15 observers are sufficient per MTurk experiment. We decided to use larger sample sizes (predetermined) for the purpose of ensuring that the statistical results are overwhelmingly convincing. Data were collected from multiple observers simultaneously (usually in batches of 15 to 30 observers), so sample sizes were chosen approximately, with final counts that depended on the number of excluded observers (see Supplementary Table 3). For the MTurk TI "Mix" dataset ($N = 475$ observers), we pooled all relevant data from a set of MTurk experiments that investigated spatial properties of the TI, to be published elsewhere (see Supplementary Method 1).

**Reporting summary**. Further information on research design is available in the Nature Research Reporting Summary linked to this article.

## Data availability
The behavioral measurements reported in the main figures are available as Supplementary Data 1. All other data are available from the corresponding author upon reasonable request.

## Code availability
The code used for data collection is available from the corresponding author upon reasonable request.

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

## Acknowledgements
This research was supported by the Basic Research Foundation, administered by the Israel Academy of Science, and by The Weizmann Braginsky Center for the Interface between the Sciences and the Humanities. We thank Drs. Misha Katkov and Noga Pinchuk-Yacobi for their suggestions and comments, and Dr. Ron Rotkopf for statistics advice.

## Author contributions
R.D. and D.S. designed, performed, and analyzed the research and wrote the paper.

## Competing interests
The authors declare no competing interests.
