## [Peer Review File · Communications Biology]

Reviewers' comments:

Reviewer #1 (Remarks to the Author):

In this manuscript, the authors report the results of several perceptual experiments, where they show that the effects of several perceptual biases varies according to the participants response times. More specifically, they report that the longer the response time, the less biases are present in the final response. They interpret their finding in the framework of sequential sampling decision models, where biases impact the starting point of evidence accumulation, explaining why their effects vanish when response time increases.

Overall, I think the manuscript does a good job in trying to account for multiple behavioral effects, observable in multiple tasks, with a single and coherent mechanism. However, I have some concerns about the analyses and claims that would need to be addressed before I can recommend this manuscript for publication.

Major point:

1) My first concern touches to the experimental design. First, the number of subjects per experiment seems at the same time quite low, and arbitrary – i.e. not determined by power analysis (e.g. 7 participants in the Discrimination (2AFC) experiment; 14 participants in the periphery TAE experiment). This raises worries about the generalizability of the results. This worry is increased by the fact that “Most observers had prior experience in participating in psychophysical experiments”. Besides, it is specified that “Most observers participated in multiple experiments of this study”, but it is unclear to which extent this is the case.

This low number of participants also seem to invalidate the use of parametric classical statistics (repeated measure ANOVAs, student t-tests, etc.).

2) I also think that the statistical analysis is inappropriate: unless RT bins were entered as parametric rather than categorical variables (which I guess was not the case, given the reported degrees-of-freedom on the F-tests), the use of repeated measure ANOVAs does not test for systematic trends in the data (e.g. increase/decrease of bias/accuracy across RT bins), which is the core class of hypotheses in the manuscript. Instead, it seems that linear (mixed-effect) regression analyses would be better suited to this purpose, and would allow to estimate absolute (and directional) effect sizes.

3) The modelling approach is a bit unorthodox and feels incomplete: models are quantitative hypotheses and should be tested against one another. Here, the null hypothesis is that model parameters should be identical in the different context. Such a model could be tested versus alternative models (which assumes different starting points, drift rates, etc...). A proper (quantitative) model comparison approach could/should be employed to determine which model best and most parsimoniously account for the observed data/biases (see e.g. Wilson and Collins. Ten simple rules for computational modeling 2019 , Palminteri and colleagues TICS 2017).

4) Also, wouldn't a DDM predict that accuracy (d') increases across RT bins? Model predictions (above and beyond the reduction in bias across RT bins) should be more comprehensively assessed, to evaluate the limits of the modelling approach.

5) I find the choice of illustrating “raw choices” (i.e. $P(\text{answer})$) in the main Figure (Fig 1.c-d; Fig 2.b-c) questionable: for instance, this raises question about whether the reduction in bias is simply due to more random choices (because they get closer to 50%). Instead, I think that the sdt approach (with bias, criterion, and accuracy) gives a better and more comprehensive account of the data (Fig S1, Fig

S3), and actually could be used to better illustrate the statistical tests (which I think should be linear regressions, and could be plotted alongside the data points - see also my point 2). I would suggest interchanging Fig 1.c-d; Fig 2.b-c with Fig S1, Fig S3, and adding regression lines...

6) It should also be noted that the main factor of interest used in the analyses is reaction times, which is an observable, rather than a true experimental condition. This can be issue, because we cannot be sure that other potential confounding factors are correctly balanced across the different levels of this factor, as it would be in a proper factorial experimental design. For example, the effects of RT could be confounded with the effects of "learning" across a block. At the very least, the author should acknowledge this limitation.

Minor:

7) Generally, I find the manuscript sometimes hard to follow. The reader is left with many holes to fill, when it comes to inferring details about the experimental design (e.g. pp.2), what the classical perceptual biases are (TAE, etc. pp.4-5), and sometimes even interpretation of some effects. Many sentences are very intricate. E.g. p.10: "Even a rapid reduction in bias (with the slower half of responses measuring practically zero bias) can be explained by a rule for when to stop accumulating evidence that depends on the extent of evidence that has been accumulated, an account with very strong support in brain decision making (...) and clear statistical implications (...).". Given the broad readership of Communication Biology, I would suggest to make sure that the manuscript remains understandable for a large audience.

8) There are typos in the manuscript. E.g. Fig.S5 p .3 discrimination (in the figure caption).

Reviewer #2 (Remarks to the Author):

The manuscript investigates across four different perceptual decision-making tasks whether the time taken to make a choice influences the bias and accuracy of choice (as defined by Signal Detection theory). They also model those decisions with the help of an evidence accumulation model and find that such a model can explain the timing effects that they found. These are important and interesting findings for the field of perceptual decision making. Despite these positives, there are several weaknesses with this paper that I detail in the following:

1) The authors need to add how the sample size for each tasks has been determined.

2) I'm also confused how the authors decided to determine the time bins. It sounds like they tried several approaches, however such a data driven approach would require a correction for multiple comparisons.

3) the same would be true for the testing of different parameters of the DDM.

4) generally for the positioning of the paper, it remains somewhat unclear what the contribution of this work is. This needs to be clarified in light of previous work.

5) Adding to this, the authors seem to generalize their work outside the realm of perceptual decision making to areas such as value-based decision-making. However, without applying their hypothesis to other decision-making tasks this remains purely speculative. This also applies to their claims on fast

vs. slow decision making, Kahneman & Tversky's (KT) work has not been established with very fast perceptual decision making in mind. Thus the critique of KT's two systems approach is difficult to be recognized in the setting of perceptual decision making tasks. In addition to that, a similar critique has already been voiced by work in decision neuroscience that has not been integrated in this paper.

6) Last but not least, the fact that one model explains the behavioral effect does not necessarily mean that there are not two (brain or psychological) systems underlying the different parameters of the model etc. Hence these claims need to be discussed more carefully in this paper, beyond what has been already done in the discussion of the current version of the manuscript on p. 10

Taken together, I like the idea of the paper, but more evidence is needed to support some of the claims made here. I would recommend the authors to pre-register their predictions and methodological approaches to replicate their initial findings and then try to either generalize them outside the area of perceptual decision-making or be more careful about how they position the paper not to generalize outside perceptual decision-making.

We thank the reviewers for their critical reading of our manuscript and for their useful suggestions. Following these comments, we rewrote sections of the paper to improve readability, and importantly, we improved the statistical analysis, we hope now to be satisfactory. In the following, we provide a detailed, point-by-point, response to the reviewers' comments.

Reply to reviewer #1:

Reviewer #1 (Remarks to the Author):

In this manuscript, the authors report the results of several perceptual experiments, where they show that the effects of several perceptual biases varies according to the participants response times. More specifically, they report that the longer the response time, the less biases are present in the final response. They interpret their finding in the framework of sequential sampling decision models, where biases impact the starting point of evidence accumulation, explaining why their effects vanish when response time increases.

Overall, I think the manuscript does a good job in trying to account for multiple behavioral effects, observable in multiple tasks, with a single and coherent mechanism. However, I have some concerns about the analyses and claims that would need to be addressed before I can recommend this manuscript for publication.

Thank you!

Major point:

1) My first concern touches to the experimental design. First, the number of subjects per experiment seems at the same time quite low, and arbitrary – i.e. not determined by power analysis (e.g. 7 participants in the Discrimination (2AFC) experiment; 14 participants in the periphery TAE experiment). This raises worries about the generalizability of the results. This worry is increased by the fact that “Most observers had prior experience in participating in psychophysical experiments”. Besides, it is specified that “Most observers participated in multiple experiments of this study”, but it is unclear to which extent this is the case. This low number of participants also seem to invalidate the use of parametric classical statistics (repeated measure ANOVAs, student t-tests, etc.).

We employ psychophysical methods that are standard in vision research. The perceptual effects studied are strong and are expected to be present in most, if not all, tested subjects. In this type of research the subjects are tested many times each, gaining experience with task, so the results are expected to be stable. Subjects can be

tested in several experiments/conditions to improve comparison between effects. This is true for experiments where prior probability of a target is manipulated (e.g. Green & Swets classical book), and for contextual effects such as the tilt-after-effect (TAE) studied here. Thus, we rely on well-established effects and methodologies. Our sample size is somewhat arbitrarily set, taking advantage of existing unpublished data collected in the lab (testing properties of TAE). The novel, and somewhat surprising, behavior reported here concerns the dependency of these effects on reaction time. For the 33 subjects in our study, there were 76 measurements across all conditions tested, with 69 measurements showing the effect (the effect was absent mainly in conditions yielding small TAE to start with, such as the non-retinotopic condition). The detailed statistical analysis described in the MS confirms the robustness of the results with the present sample size, showing no need to add subjects.

We indeed use parametric tests. We conferred with a statistician about non-parametric alternatives, and to our understanding, the main relevant alternative is using a Spearman regression analysis on the raw trial data (i.e., correlating +/- answers with RT), but this, being confounded with recognition accuracy, does not directly test the property of interest (bias). We therefore prefer to use parametric statistics (which are more simple and straightforward to interpret), and supplement with non-parametric statistics (which we now employ in the model comparison) and by also providing individual data (which we now show for all experiments).

2) I also think that the statistical analysis is inappropriate: unless RT bins were entered as parametric rather than categorical variables (which I guess was not the case, given the reported degrees-of-freedom on the F-tests), the use of repeated measure ANOVAs does not test for systematic trends in the data (e.g. increase/decrease of bias/accuracy across RT bins), which is the core class of hypotheses in the manuscript. Instead, it seems that linear (mixed-effect) regression analyses would be better suited to this purpose, and would allow to estimate absolute (and directional) effect sizes.

Following the reviewer's comment we changed the statistical analysis to linear mixed-effect regression analysis instead of repeated-measures ANOVA. This analysis showed about the same level of statistical significance for all effects. Note that we used the bin index as the regressor because RT distributions are skewed.

As to effect sizes, we now report Cohen's d and Hedges' g between the first and the last bins, finding ~3 for the prior-dependent bias experiments, and ~1 for the context-dependent experiments (see in text).

3) The modelling approach is a bit unorthodox and feels incomplete: models are quantitative hypotheses and should be tested against one another. Here, the null hypothesis is that model parameters should be identical in the different context. Such a model could be tested versus alternative models (which assumes different starting points, drift rates, etc...). A proper (quantitative) model comparison approach could/should be employed to determine which model best and most parsimoniously

account for the observed data/biases (see e.g. Wilson and Collins. Ten simple rules for computational modeling 2019 , Palminteri and colleagues TICS 2017).

We wish to stress that our main point is not related to a specific model, but to a general theoretical point that can be implemented in different models. Therefore, in our opinion, model comparison is a secondary consideration. That being said, we have added a detailed model comparison using the AIC and BIC measures. We are happy to see that the outcome clearly confirms our claims.

We note that the null hypothesis we used is stricter than the one proposed by the reviewer. Specifically, the reviewer proposed using the null hypothesis that model parameters are identical in the different contexts. According to this hypothesis, there is no context-dependent bias, i.e., no TAE or TI. In our opinion, this null hypothesis is too weak (and not relevant): the question considered here is concerned with the interaction between TAE/TI and RT. Therefore, we used a null hypothesis of fixed bias across RTs (which is equivalent to a bias that reflects a change in the drift rate in the DDM).

4) Also, wouldn't a DDM predict that accuracy (d') increases across RT bins? Model predictions (above and beyond the reduction in bias across RT bins) should be more comprehensively assessed, to evaluate the limits of the modelling approach.

This is somewhat not intuitive, but d' is not predicted to change with time in the standard (basic) DDM. Note that in DDM temporal integration is bounded, with the bounds defining a fixed level of accumulated evidence, hence a fixed value of d' . (This may be more intuitive considering the SPRT interpretation of DDM, in which the bounds are interpreted as alpha and beta error thresholds, Wald 1945.) We of course acknowledge that this (unintuitive) point needs to be mentioned in text (see page 10-11). But again, we try here to avoid a detailed model analysis (many variants, too many parameters) and emphasize a universal property of noisy evidence accumulation.

5) I find the choice of illustrating “raw choices” (i.e. $P(\text{answer})$) in the main Figure (Fig 1.c-d; Fig 2.b-c) questionable: for instance, this raises question about whether the reduction in bias is simply due to more random choices (because they get closer to 50%). Instead, I think that the sdt approach (with bias, criterion, and accuracy) gives a better and more comprehensive account of the data (Fig S1, Fig S3), and actually could be used to better illustrate the statistical tests (which I think should be linear regressions, and could be plotted alongside the data points - see also my point 2). I would suggest interchanging Fig 1.c-d; Fig 2.b-c with Fig S1, Fig S3, and adding regression lines...

Done, for the prior-dependent figure. Not for the context-dependent figure (now Supplementary Figure 5), which we believe offers little beyond repetition of the same

control (d') in a different condition. We avoided regression lines, because the bounded/unbounded models do not predict a linear decrease.

6) It should also be noted that the main factor of interest used in the analyses is reaction times, which is an observable, rather than a true experimental condition. This can be issue, because we cannot be sure that other potential confounding factors are correctly balanced across the different levels of this factor, as it would be in a proper factorial experimental design. For example, the effects of RT could be confounded with the effects of “learning” across a block. At the very least, the author should acknowledge this limitation.

The general point raised by the reviewer is now acknowledged in the text.

The relevant text excerpt (p. 11):

“The factor we use for the analyses, RT, is an observable rather than an experimentally manipulated factor. This design decision raised the need to control for potential confounding factors (as discussed in the text), but importantly, note that an observable cannot always be replaced by an experimentally manipulated factor. Notably, under the assumptions of the DDM, the stochasticity in diffusion underlies co-variability of RT and bias. Therefore, considering the slower/faster subset of trials (as we do here) is not immediately interchangeable with an experimental manipulation that affects the average RT (by changing the separation between the bounds).”

We also added a control analysis which rules out learning as a confound for the prior-dependent experiments where it is most relevant (Supplementary Fig. 3).

Minor:

7) Generally, I find the manuscript sometimes hard to follow. The reader is left with many holes to fill, when it comes to inferring details about the experimental design (e.g. pp.2), what the classical perceptual biases are (TAE, etc. pp.4-5), and sometimes even interpretation of some effects. Many sentences are very intricate. E.g. p.10:

“Even a rapid reduction in bias (with the slower half of responses measuring practically zero bias) can be explained by a rule for when to stop accumulating evidence that depends on the extent of evidence that has been accumulated, an account with very strong support in brain decision making (...) and clear statistical implications (...).”

Given the broad readership of Communication Biology, I would suggest to make sure that the manuscript remains understandable for a large audience.

An effort was made to improve readability.

8) There are typos in the manuscript. E.g. Fig.S5 p .3 discrimination (in the figure caption).

Fixed.

Reply to reviewer #2:

Reviewer #2 (Remarks to the Author):

The manuscript investigates across four different perceptual decision-making tasks whether the time taken to make a choice influences the bias and accuracy of choice (as defined by Signal Detection theory). They also model those decisions with the help of an evidence accumulation model and find that such a model can explain the timing effects that they found. These are important and interesting findings for the field of perceptual decision making. Despite these positives, there are several weaknesses with this paper that I detail in the following:

Thank you!

1) The authors need to add how the sample size for each tasks has been determined.

See the response above to reviewer #1 (point 1) concerning this issue. Our sample size is somewhat arbitrarily set, taking advantage of existing unpublished data collected in the lab (testing properties of TAE). There were at least seven participants per experiment, which we believe is representative and sufficient for the purpose of studying basic visual phenomena using within-individual effects that are robust in the sampled population.

2) I'm also confused how the authors decided to determine the time bins. It sounds like they tried several approaches, however such a data driven approach would require a correction for multiple comparisons.

We assume that the reviewer refers to the number of time bins considered, which was 2, 4, or 6. Using 2 bins is obviously too limiting to investigate dynamics, but we acknowledge that our decision for using 4 vs. 6 bins depended on the data (specifically, the decrease in context-dependent bias was slower, so we used 6 bins). We now correct for two multiple comparisons in all relevant statistics. Regardless, we wish to emphasize that the findings were extremely robust to the analyses approach. Most importantly, the conclusions from the statistical analysis do not depend on the RT binning, i.e., are always significant (always $p \leq 0.002$ for reduction in bias with RT, corrected for two multiple comparisons).

Other analysis decisions were motivated by best practice/necessity, so they did not inflate the alpha. The main idea we followed was to be as strict as possible, to prevent/minimize confounds. (Trying a confounded alternative is useful as a sanity check. Specifically, ignoring the prior/context in binning leads to perfect alignment of the time bins, see slight misalignment in Supplementary Figs. 1, 4. We wanted to verify that the results are not contingent on the chosen analysis.)

3) the same would be true for the testing of different parameters of the DDM.

We now perform a detailed model comparison, in which we take into account the number of considered models.

4) generally for the positioning of the paper, it remains somewhat unclear what the contribution of this work is. This needs to be clarified in light of previous work.

Main contributions:

- Observing that context-dependent perceptual bias is decreased with longer RT.
- Explaining reduction in bias with decision time using a single stochastic system compatible with standard neuronal modelling.

We note that the observations reported here are novel and non-trivial, in particular regarding the contextual effects. The TAE is known to depend on target duration, but this was explained in reference to stimulus time, not to decision time. Furthermore, these duration effects have no quantitative accounts, but rather are verbally explained by different levels of processing.

5) Adding to this, the authors seem to generalize their work outside the realm of perceptual decision making to areas such as value-based decision-making. However, without applying their hypothesis to other decision-making tasks this remains purely speculative. This also applies to their claims on fast vs. slow decision making, Kahneman & Tversky's (KT) work has not been established with very fast perceptual decision making in mind. Thus the critique of KT's two systems approach is difficult to be recognized in the setting of perceptual decision making tasks. In addition to that, a similar critique has already been voiced by work in decision neuroscience that has not been integrated in this paper.

Right, the generalization of our theory beyond perceptual decision making is speculative, but we believe it to be an interesting one. We definitely do not present a critique of KT, and of course acknowledge there may exist decision-making tasks that are mediated by dual brain systems (p. 14). Our critique is for the generalization of the two-system approach to perceptual decisions (see our response to point 4), showing that a well-defined, simple, quantitative model can account for the data. Our work provides an alternative account for perceptual/behavioral effects now explained by Type-I and Type-II processes. To the best of our knowledge, the effects reported here were not reported before (see point 4 above).

6) Last but not least, the fact that one model explains the behavioral effect does not necessarily mean that there are not two (brain or psychological) systems underlying the different parameters of the model etc. Hence these claims need to be discussed

more carefully in this paper, beyond what has been already done in the discussion of the current version of the manuscript on p. 10

Sure, we agree, see above and in the text.

The relevant text excerpt (p. 14):

“Of course, there can be dual processes in the brain, and there are low vs. high brain areas, but at least for the basic perceptual phenomena considered here, there is a simple mechanistic account that does not need to assume multiple systems. Note that it is still possible that the different model parameters are set by two or more brain or cognitive modules. For example, the starting point (prior) is possibly set by a higher-level experience-based module, which takes into account information over a long time, while the information accumulation is possibly performed by a lower-level sensory-based module. The specific anatomical analogue may depend on the behavioral task at hand.”

Taken together, I like the idea of the paper, but more evidence is needed to support some of the claims made here. I would recommend the authors to pre-register their predictions and methodological approaches to replicate their initial findings and then try to either generalize them outside the area of perceptual decision-making or be more careful about how they position the paper not to generalize outside perceptual decision-making.

Thanks, we are considering extensions.

Reviewers' comments:

Reviewer #1 (Remarks to the Author):

The authors have, for the major part, addressed my previous comments. Again, although I feel quite convinced by the results and quite sympathetic with the authors conclusions, I still feel that some steps should be taken to ensure that this work is understandable and reproducible.

1- Regarding the sample size, the author should still specify in the manuscript how it was determined. If the manuscript is targeted at a broad audience, researchers from some fields (e.g. behavioral economists) would never read a paper with such a small sample size, and no justification for it. If the author claim that, from past literature, the effects of interest are extremely robust and justify a small sample size to observe them with sufficient power, this should be explicitly stated (and, ideally, quantitatively assessed).

2- Again, for a potential readership that is unfamiliar with DDM, I think the method section should contain the mathematical specification of the different variants of the DDM used in the manuscript.

3- The choice of a fixed-effect model (i.e. one set of parameter for the whole population) for the DDM fit/model comparison seems very unorthodox. Do the results hold if one set of parameter is fitted per individual?

Reviewer #2 (Remarks to the Author):

The revised manuscript "Seeing, fast and slow: the effects of processing time on perceptual bias" has addressed most of my concerns.

For 1) and 2) it would be helpful for readers and to comply with current good standards of science to explain the sample size and analysis choices in a foot note or appendix.

Well done!

We thank the reviewers for their comments and suggestions. We followed all suggestions, and successfully replicated our findings through the Amazon Mechanical Turk with a large number of participants. We hope the reviewers find the results as convincing as we do.

Follows is a point-by-point reply. The major textual changes in the manuscript files were colored red.

Reviewer #1 (Remarks to the Author):

The authors have, for the major part, addressed my previous comments. Again, although I feel quite convinced by the results and quite sympathetic with the authors conclusions, I still feel that some steps should be taken to ensure that this work is understandable and reproducible.

1- Regarding the sample size, the author should still specify in the manuscript how it was determined. If the manuscript is targeted at a broad audience, researchers from some fields (e.g. behavioral economists) would never read a paper with such a small sample size, and no justification for it. If the author claim that, from past literature, the effects of interest are extremely robust and justify a small sample size to observe them with sufficient power, this should be explicitly stated (and, ideally, quantitatively assessed).

Done.

The claim that the effects under investigation are expected to be extremely robust is now explicitly stated and referenced (see “sample size” in pp. 29-30). Some of the effects were initially estimated using existing data from our lab previously not analyzed for reaction time.

Most importantly, the results from a huge population of hundreds of subjects are now presented. The data, collected using the Amazon Mechanical Turk platform, showed a highly significant replication of our findings. Almost all figures were updated, see e.g. Figs. 2 and 3, and the statistical summary in Supplementary Table 1.

We also improved the analysis in three ways. (i) For the context-dependent experiments, we now use four data bins instead of six. This was done to improve the measurement range in the TI experiments, which previously saturated the range. (We correct for multiple comparisons.) (ii) For the context-dependent experiments again, bias is now measured using data from the two target orientations closest to the "perceived" vertical orientation, where applicable (see pp. 24). This change accommodates observers having perceived orientation that is non-vertical (e.g., due to a slightly tilted display), and further improves the dynamic range. We emphasize that

the new Mechanical Turk results are not contingent on the improved analysis, as evident in Supplementary Fig. 7 where data for the physical vertical orientation is presented. (iii) The fitting function was improved so more optimal parameters can be found. This led to slightly improved AICc/BIC scores for all models, and also, a more heavy reliance on the measurement range when parameters are fitted (see Fig. 6B).

2- Again, for a potential readership that is unfamiliar with DDM, I think the method section should contain the mathematical specification of the different variants of the DDM used in the manuscript.

We now describe the mathematical formulae which we use in the fit (see pp. 25-27, and in particular equations (5) and (6)).

3- The choice of a fixed-effect model (i.e. one set of parameter for the whole population) for the DDM fit/model comparison seems very unorthodox. Do the results hold if one set of parameter is fitted per individual?

Fixed.

We updated the model comparison to use mixed-effects models having one fitted parameter per observer (corresponding to the magnitude of individual bias). Additional group-level parameters are used where relevant (see pp. 25-27). The results are overwhelmingly in support of main point (i.e., against the null-hypothesis “Fixed” model, see Table 1). We wish to stress, as before, that our main point is not related to a specific model, but to a general theoretical idea that can be implemented using different models. As such, we acknowledge that the data cannot convincingly discriminate between the alternatives to the null hypothesis (bounded vs. unbounded), or between different definitions of the group-level parameter (we used the ratio of starting point to drift rate changes, which seems to be the most agnostic definition).

Reviewer #2 (Remarks to the Author):

The revised manuscript “Seeing, fast and slow: the effects of processing time on perceptual bias” has addressed most of my concerns.

For 1) and 2) it would be helpful for readers and to comply with current good standards of science to explain the sample size and analysis choices in a foot note or appendix.

Well done!

Thank you.

Analysis choices are now better explained throughout the Methods section. The sample size is now explained in a separate section of the Methods (pp. 29-30).

REVIEWERS' COMMENTS:

Reviewer #1 (Remarks to the Author):

In this revised manuscript, the author added extensive web-based experimental replications of their main findings. The main result - namely a reduction of decision biases with increasing RTs for a large portfolio of perceptual tasks - appears very robust and replicable (the supplementary tables 1-3 give an impressive overview of the robustness of the findings).

The authors also addressed my comments about mixed/random-effects in modelling, and made extensive updates in the methods sections to clarify the DDMs.

I am happy with the revisions, and think that the manuscript is suitable for publication.

I also want to congratulate the author for this very interesting and impressive piece of work (the generalizability and replicability of the result is quite spectacular), and thank them for their constructive attitude during this review process.

Reviewer #2 (Remarks to the Author):

The authors have improved the manuscript in important ways. It is very valuable that they collected data from a larger sample size using M-Turk. I understand that in their field the authors usually work with smaller sample sizes since they have lots of data points in individuals by applying within-participant repeated measure designs. However, an N=50 is not extraordinary large! Again, I do think it is, of course, sufficient for the goal of this paper given the large effect sizes BUT one last request I would have is that the authors replace the somewhat misleading sentence "replicated with a large sample size" line 87 with "replicated with a larger sample size" (same for other references to the m-turk pool). This is because one could argue from a social science perspective 50 m-turk participants are still small. This is because m-turk data is noisy as compared to lab data and also RT have noise because the measure could be confounded by random differences in internet connection speed.

The authors also need to add how they determined the sample sizes for the different m-turk studies since they vary a lot (50- over 400) and how many participants were dropped for each of the listed a-priori defined exclusion criteria per study.

Following the editorial request, we have changed all figure panel letters from uppercase (A/B/C) to lowercase (a/b/c), and have made the minor changes requested by the second reviewer, as described below. (The reviewer's comments are in blue.)

The authors have improved the manuscript in important ways. It is very valuable that they collected data from a larger sample size using M-Turk. I understand that in their field the authors usually work with smaller sample sizes since they have lots of data points in individuals by applying within-participant repeated measure designs. However, an N=50 is not extraordinary large! Again, I do think it is, of course, sufficient for the goal of this paper given the large effect sizes BUT one last request I would have is that the authors replace the somewhat misleading sentence "replicated with a large sample size" line 87 with "replicated with a larger sample size" (same for other references to the m-turk pool).

Fixed all such references.

This is because m-turk data is noisy as compared to lab data and also RT have noise because the measure could be confounded by random differences in internet connection speed.

To clarify, we use a more sophisticated technology, RTs were measured locally on the MTurk worker machine, so internet connection speed does not affect the measured RT; this clarification was added to lines 430-431 (line numbering refers to the word document with Track Changes set to "No Markup").

The authors also need to add how they determined the sample sizes for the different m-turk studies since they vary a lot (50- over 400) and how many participants were dropped for each of the listed a-priori defined exclusion criteria per study.

Added, see lines 774-782 in the main text and Supplementary Table 3.